# Is the Concept of Solastalgia Meaningful to Pacific Communities Experiencing Mental Health Distress Due to Climate Change? An Initial Exploration

**DOI:** 10.3390/ijerph20227041

**Published:** 2023-11-09

**Authors:** Trish Tupou, Jemaima Tiatia-Siau, Christina Newport, Fiona Langridge, Suelaki Tiatia

**Affiliations:** 1Crawford School of Public Policy, Australian National University, Canberra, ACT 2601, Australia; patricia.tupou@anu.edu.au; 2School of Māori Studies and Pacific Studies, University of Auckland, Auckland 1010, New Zealand; c.newport@auckland.ac.nz (C.N.); suelaki.tiatia@auckland.ac.nz (S.T.); 3Paediatrics, Child and Youth Health, University of Auckland, Auckland 1023, New Zealand; f.langridge@auckland.ac.nz

**Keywords:** climate change, indigenous, Pacific peoples, mental health distress, solastalgia, wellbeing

## Abstract

The critical inquiry is how Pacific communities themselves characterize mental distress as a result of climate change. If not solastalgia, what more suitable terms might they use? This viewpoint article aims to initiate a discourse using solastalgia as the focus for the Pacific by 1. providing a definition of solastalgia; 2. examining its application in Pacific research; 3. presenting limitations of solastalgia; and 4. assessing its appropriateness for Pacific communities. There is a dearth of research using solastalgia, particularly within Pacific communities. The Pacific region’s diverse contexts may already possess terms that effectively convey place-based distress that solastalgia attempts to describe. However, the authors found that solastalgia holds limited utility in the Pacific region, primarily based on a review of the literature, which involved keyword searches in Google Scholar such as solastalgia, mental health, mental distress, wellbeing, climate change, environmental distress, displacement, and Indigenous and Pacific peoples. More importantly, the concept is limited in capturing Pacific experiences of land loss due to climate change events, particularly, as the Pacific imbues land with profound significance, intertwined with culture, identity, and wellbeing. Land loss equates to a loss of culture, identity, wellbeing, and kinship in most Pacific contexts. It is apparent that broader and more holistic approaches are required.

## 1. Introduction

Mental health impacts of global climate change have been documented in various empirical studies and systematic reviews, yet the understanding of the full scope of these impacts remains limited [1], particularly for Pacific peoples. Moreover, from across the Pacific region, a place highly susceptible to adverse climatic events, there is an invisibility of the lived experiences and voices of Pacific peoples [2,3]. Our viewpoint draws on a comprehensive analysis of the existing literature, which entailed conducting keyword searches in Google Scholar using terms such as solastalgia, mental health, psychological distress, wellbeing, climate change, environmental distress, displacement, and Indigenous and Pacific peoples.

A formidable voice, among many from across the Pacific region, is that of Marshallese climate change activist, poet, and performance artist—Kathy Jetñil-Kijiner. In her poem “Dear Matafele Peinam”, an ode to her daughter, she describes the imperial injustices and an awareness raising of her country’s rising sea waters. A Pacific voice indigenous to the Republic of the Marshall Islands is powerfully conveyed in the following excerpt:
*Dear matafele peinam,**I want to tell you about the lagoon**that lucid, sleepy lagoon lounging against the sunrise**men say that one day**that lagoon will devour you**they say it will gnaw at the shoreline**chew at the roots of your breadfruit trees**gulp down rows of your seawalls**and crunch your island’s shattered bones**they say you, your daughter**and your granddaughter, too**will wander rootless**with only a passport to call home*[4]

Jetñil-Kijiner brings to light the manner in which climate change impacts particular peoples and locations, yet on a global platform, is essentially invisible. She imagines a forthcoming era wherein her nation is submerged, and her daughter is devoid of a residence; she underscores the ineffectiveness of policymakers and worldwide influencers in addressing the ramifications of climate change. Despite this, the passport endures, even as the land it symbolizes has disappeared [5].

### Place Identity and Mental Health in the Pacific

As well as accounting for distinctions between relationships to land for Pacific and non-Pacific peoples, there is also a need to differentiate between the way we define mental health and wellbeing. Solastalgia works from Western definitions of wellbeing; however, researchers may find that such definitions do not necessarily resonate with Pacific communities. For instance, as Aotearoa New Zealand’s Government’s Mental Health and Addiction inquiry report, He Ara Oranga [6], asserts:

“*Aiga/koputangata/kāinga/magafaoa/matavuvale/kāiga (family) is central to Pacific mental health and wellbeing, including family support and inclusion in decision-making*” [6] (p. 87). Relationships to the land are also convoluted by this tethering together of self and place. As has also been outlined in He Ara Oranga: Report of the Government Inquiry into Mental Health and Addiction
*For Pacific peoples, mental health and wellbeing encompasses a holistic approach of reciprocity, respect, belonging, genealogy, and relationships with all entities—Atua, the land and environment, ancestors, cultures, languages, family and others, collectivism—elements that protect and strengthen family and individual wellbeing*[6] (p. 22)

The environment, place identity, and wellbeing cannot be separated within Pacific contexts, and it remains unclear where, and how, solastalgia will account for these varying and complex understandings of place and place-based distress.

The term ‘solastalgia’ was developed by environmental philosopher, Glenn Albrecht, in 2005 due to a need to name the growing phenomenon of distress experienced by those who remain in places/lands going through drastic environmental transformation. Albrecht coined the term solastalgia to encompass the distress related to environmental change affecting one’s health by which:
*“Solastalgia is the pain or sickness caused by the loss or lack of solace and the sense of isolation connected to the present state of one’s home and territory”*[7] (p. 45)

Originally, this was used to describe the experiences of the people in the Upper Hunter Valley, New South Wales, Australia, a prominent open-cut mining town. As Albrecht found, the individuals he expressed concern for were not facing forcible displacement from their homes or lands. Nonetheless, their distress was related to a sense of powerlessness and a perception of environmental injustice being imposed upon them [7]. A connection was forged between place and the melancholia “*connected to lack of solace and intense desolation*” [7] (p. 44). Albrecht also made the distinction between the differing feelings of solastalgia felt between Indigenous place-based identities and other place-based identities. He contended that for Indigenous peoples, such a feeling of nostalgia and solastalgia is ongoing, and the loss of ancestral lands and the erosion of cultural connections between Indigenous communities and their land, which can be seen as the foundations of their identity, are implicated in various dimensions of the ongoing ‘crisis’ experienced by many Indigenous communities in present-day Australia [7]. There has not yet been a clarification of why Indigenous communities, and, for the purposes of this discussion, Pacific communities may experience solastalgia differently to those who create place-based identities based on their work, livelihoods, or generational connection to certain landscapes. However, the second-named author and colleagues are currently examining, more explicitly, the concept of solastalgia as conceptualised by Pacific study participants themselves, as part of a larger Health Research Council of New Zealand-funded study: *Climate Change and Mental Wellbeing: The Impact on Pacific Peoples* [3], which focuses on Niue, the Cook Islands and Christchurch, Aotearoa New Zealand. 

Solastalgia may be classified as a “psychoterratic” state, as it endangers people’s mental health by disrupting the harmonious bond between individuals and their homes or land. It incorporates elements of nostalgia, where the yearning for home arises not from being physically distant, but from longing for a specific idealized notion of home while still residing there. This condition is induced by alterations to the natural environment, such as floods, droughts, land degradation, or the depletion of cherished ecosystems [7]. In Table 1, Albrecht’s Tenets of Solastalgia are outlined. 

Migration can serve as either a proactive measure or a last resort in addressing climate change. Even so, prior to migration taking place, there exists the possibility of experiencing mental distress due to the ongoing deterioration of environmental conditions. Consequently, in both scenarios, it is anticipated by the authors that a significant number of individuals residing in the Pacific region will undergo some degree of distress as the impacts of climate change intensify and impact those remaining on the islands. As mentioned, an innovative Health Research Council of New Zealand-funded study: *Climate Change and Mental Wellbeing: The Impact on Pacific Peoples* [3], led by the second-named author, tested the relevance of the notion of ‘solastalgia’ for the purposes of exploring the link between climate change and mental health and wellbeing in the Pacific. In phase one of the study [3], panellists confirmed Pacific perspectives of the impacts of climate change on mental wellbeing, primarily related to the loss of land and livelihoods. Of particular importance was losing connection to ancestors and history, as well as nurturing of resilience factors unique to Pacific peoples such as connection to culture and family. Phase two findings reinforced these perspectives, and the results are forthcoming.

The critical question is how do Pacific peoples themselves define this type of mental health distress? If not solastalgia, what are the more appropriate expressions? The aim of this viewpoint article is to introduce a discussion on solastalgia in relation to Pacific peoples by 1. defining solastalgia; 2. investigating the use of solastalgia in Pacific research; 3. outlining the critiques and limitations of solastalgia; and 4. exploring the appropriateness of this term for Indigenous and Pacific populations.

## 2. Solastalgia in Research

### Solastalgia and the Pacific

To date, there has been no solastalgia research conducted *by* or *with* Indigenous peoples [8], as is the case for Pacific communities; thus, exploring this term as a conceptual framework is novel for Pacific contexts, particularly if the research is predominantly undertaken by Pacific peoples. With the exception of a short article by McNamara and Westoby [9] on the potential gendered nature of climate change amongst Erub Islander women in the Torres Strait, there is no research on solastalgia and the Pacific or Pacific peoples. This is not surprising considering there has been minimal research undertaken on climate change in conjunction with mental health and wellbeing in the Pacific in general. Nevertheless, Hunter [10] highlighted the compounding effect that climate change will pose on the mental wellbeing of Indigenous peoples in Australia, including Torres Strait islanders, who already experience adverse mental health challenges and outcomes. There is emerging research on climate change and mental health in the Pacific [2] that outlines research on this topic with a specific focus on mental health outcomes following natural disasters. Furthermore, Howard-Chapman et al. [11] also discussed the potential for mental distress among Pacific communities who will undergo climate-induced migration to destinations like Aotearoa New Zealand. Pacific peoples in Aotearoa New Zealand also experience higher rates of mental health challenges compared with the general population [6]. Therefore, as climate change forces migration, there is a concern that these individuals may encounter exacerbated adverse mental health concerns due to the combination of pre-existing challenges and the stressors associated with displacement and adaptation to a new environment.

## 3. Limitations of Solastalgia

### 3.1. Theoretical

There are significant conceptual blind spots in solastalgia-focused research specifically for Pacific peoples. For instance,
The term itself does not consider a broader structural analysis, by focusing on individual and community relationships to land and place identity within predominantly affluent and settler communities. Moreover, embedded socio-political structures that mediate such relationships are not necessarily considered. With such a focus, there is the potential for structural inequities to be heightened or overlooked. This is especially salient for health-related research with Pacific communities who have been largely absent from solastalgia research.However, the current evidence is clear that initial recognition of the stress and anxiety caused by climate change will be significant in national policy development, particularly in the expansion of current mental health distress categories and deeper understanding regarding Pacific peoples, more specifically.Another limitation is the absence of defining land and place centred around Pacific peoples’ perspectives. Research utilising the term solastalgia tends to focus on a Western understanding of place, or if acknowledging other worldviews, these are not fully described from *local* perspectives. As an example, McNamara and Westoby’s [12] paper on Erub Islanders from the Torres Strait stands out. In their interviews with four local aunties, they apply solastalgia as a way to connect their experiences with others who are at the frontlines of climate change. The purpose seems unclear as to what function or use such a term can produce. However, in an earlier paper by McNamara and Westoby [9], they seek to look at the importance of understanding local knowledge with regard to climate change research.As previously mentioned, considerations of the land and how this is defined are often not found in solastalgia-related research, so it is interesting that these orientations were not discussed in collaboration with this term and their other work. In an excerpt from their interviews with elders, one elder mentions the reading of the frigate bird—that a change in bird migration signals a change to their environment [12] (p. 896). This is important in highlighting how the term solastalgia may be experienced differently by those who inherently hold a local worldview. Perhaps there is a need for such a term to be adapted to extend beyond a concept that relates to nostalgia for the state of the land as it once was but also in the ways that land or space can be read and understood: that the interactions with land are changed and felt differently by Pacific peoples. If such a consideration is taken into account, it is highly likely that more appropriate local terms will come to the fore.Further, Tschakert and Tutu [13] also argued that solastalgia does not capture other stressors outside of climate change/environmental destruction that contribute to cumulative distress. This is important when considering Pacific contexts as other factors such as socio-economic positioning, remoteness, colonisation, and other socio-political stressors may inform the distress caused by climate change. This is also noted by Askland and Bunn [14], who state that perhaps a more holistic approach to place-based distress is necessary to more fully comprehend the impact that climate change poses to particular ‘vulnerable’ communities.

### 3.2. Pacific Research

We tested the use of solastalgia in both a review of the literature and interviews with Pacific study participants (findings forthcoming) and found that there are significant limitations to utilizing the concept of solastalgia for research and work in the Pacific. Climate change and health research in the Pacific has been largely conducted from the outside to the inside by people not from or intimately connected to these lands, resulting in a mismatch between varying approaches to research. Hayward et al. [15] argue against prominent deficiency-focused discourse, highlighting that “*new generations, strengthened by local values, and new advances in scientific knowledge, also bring new possibilities for effective climate action*” (p. 5). It is important to reflect on how solastalgia might fit into this call for local values and how the Pacific’s experience with climate change may provide new conceptualisations of the impact of the changing environment on one’s mental health distress.

In their scoping review of solastalgia, and of relevance to Pacific peoples, Galway et al. [8] suggest that future work of this nature needs to “… better understand Indigenous peoples’ lived experiences of landscape transformation and degradation in the context of historical traumas” [8] (p. 2677). Solastalgia work has not considered the socio-political context of relationships to land and how these are forged, maintained, and changed over time. To conceptually underpin climate change and mental health distress research in the Pacific without an acknowledgement of the socio-political formations that govern this region, would be to ignore the broader macro structures that have created the need for such research in the first place. 

Relevant to our *Climate Change and Mental Wellbeing: The Impact on Pacific Peoples* [3] study mentioned earlier, and thoughts on migration, is Tschakert and Tutu’s [13] work on the correlation between environmental distress and migration. They posit that the degree of distress an individual or a community experiences is intricately linked to the loss of their deep-rooted connection to a specific place. In their study on the internal migration of Northern Ghana communities, they utilised the Environmental Distress Scale (EDS) and Likert Scale to assess solastalgia based on participants’ perceptions of distress and ‘thresholds for pathological homes’ in tandem with semi-structured interviews with residents of affected rural villages. Their findings revealed that the motivations behind migration are often multifaceted, making it challenging to determine whether migration as a whole or, more specifically, factors like the pursuit of arable land for farming or addressing social conflicts serve as the primary response to threats to their livelihoods [13].

Further investigation into the links between climate change, migration, and solastalgia research would be beneficial to capture the complexities of (im)mobility and movement within the region, with the potential of connecting these future shifts to historical patterns of movement. 

### 3.3. Climate-Induced Migration and Mental Health Distress

Beyond solastalgia, one should also consider the mental health distress effects of climate change on the habitability of one land and potential displacement, relocation, and migration within and across international borders, particularly to more urbanised areas. As a term used to describe the specific distress experienced by those who *stay*, it is unsurprising that research associated with this term does not have a strong focus on those who leave. 

In some cases, migration has improved one’s overall mental health and wellbeing; however, this has mostly been in relation to building stronger social ties and financial support in the places that one migrates to [16]. Under certain conditions—such as climate-induced migration—we may expect a different set of outcomes given the nature of this move and the possibility of not being able to return. Regardless, more research that examines solastalgia and migration needs to be undertaken as this is currently largely absent from the literature, with the exception of Tschakert and Tutu [13], as previously discussed. Given this, it is not clear if the term solastalgia is suitable for migration research; however, the threat of migration may be relevant to environmental distress and triggers. 

### 3.4. Solastalgia and Indigeneity

There is little work that uses solastalgia both by and for Indigenous communities [8] and even less work that speaks directly to Pacific communities. This may suggest that Indigenous/Pacific researchers and/or Indigenous/Pacific communities do not relate to this type of research, or that perhaps, other terminology is preferred. Contexts in the Pacific vary and may have their own terms that already illustrate the place-based distress that solastalgia seeks to diagnose and define. 

According to Cunsolo Willox et al. [17], and of relevance to Pacific contexts, climate change disproportionately affects Indigenous peoples. In their paper, which alludes to the Indigenous people of Nunatsiavut, Canada, they note that the physical changes to the land disrupt the capacity of the Inuit to sustain their culturally and socially significant land-based activities. This is an important distinction that is largely missing from solastalgia research—that land is not just used for capital, work-based utility, or even as holders of family histories for inter-generational settler farming communities. For Pacific peoples, for instance, land may also act as a facilitator of cultural and social activities, and in many parts of the Pacific, land is considered an ancestor [18,19,20], whilst in others, land has its own genealogies and agency [19]. These distinctions may not be fully captured or realised through solastalgia research, which has its foundations in settler communities and has largely continued to work in affluent discourses. Work in the Pacific may require the fleshing out of solastalgia to provide for these potential relationships with land. Alternatively, other terminology and/or conceptual frameworks may be more fitting. 

Our findings from our *Climate Change and Mental Wellbeing: The Impact on Pacific Peoples* [3] project demonstrate that the implications of climate change on Pacific peoples’ overall health outcomes are being felt. We discovered that solastalgia as a concept may be of limited value in most Pacific contexts. Our research highlights that it fails to encapsulate Pacific experiences of land loss due to climate change events. The worldviews of the Pacific peoples endow land with sanctity and a deep connection to culture, identity and wellbeing. For Pacific communities, the loss of land is ultimately a loss of culture, identity, wellbeing, and kinship. Therefore, the impact of climate change on Pacific peoples is economic, mental, physical and spiritual. On this premise, the concept of solastalgia may be narrow in its ability to fully capture and convey the complex realities experienced by Pacific peoples in relation to land displacement.

According to preliminary findings in our research, the oral traditions of the Cook Islands recount the tale of involving Ati, a husband, and his wife Mōmoke, who hailed from the underworld. Mōmoke, desiring a reunion with her family and for them to meet their son, made the decision to dive into a deep pool. Holding their son, Ati attempted to follow, but his inability to hold his breath prevented him from reaching the depths. Despite multiple efforts, he ultimately relinquished his attempts and sat beside the pool grieving for Mōmoke, fully aware that she would never return. In honour of their separation, he named their son Ati’ve (meaning separation) and sealed off the pool.

The mourning of separation in the legend of Ati and Mōmoke illustrates the mourning of separation Pacific peoples currently face as a result of the loss of their ancestral lands. As described by Enari & Jameson [21], Pacific peoples consider their ancestral homelands as their (Mother)land. Thus, in Pacific contexts, the loss of land is comparable to the mourning of the loss of kin. It is evident that although the concept of solastalgia sheds light on the distress people experience as a result of environmental transformation, much of the literature on solastalgia is limited in describing Pacific experiences.

As mentioned, most of the solastalgia-related evidence has focused on Australian landscapes and settler communities, thus reflecting its origins. Moreover, research has applied this term as both a theoretical concept and as a diagnosis of environmental distress. This means that in large, solastalgia is *applied* retroactively and its function remains as such.

After solastalgia’s inception [7], Albrecht et al. [22] conducted further research using this concept with people in the Upper Hunter area of New South Wales, Australia. They found that residents of the Upper Hunter Valley were experiencing solastalgia in response to the open-cut mining and power station fallout in their local community. Drawing on the pathology of nostalgia, Albrecht et al. argued that climate change is likely to be one of the drivers behind “nostalgia as a serious form of pyschoterratic illness” [22] (p. 96) in the twenty-first century. As an extension of this, solastalgia is distinct from nostalgia as it is used to explain the experience of not being ‘displaced’ by climate change but instead having to stay within the conditions and witness the environmental deterioration of ‘home’ and therefore feeling nostalgic for a past state of place. Residents of the Upper Hunter area who witnessed open-cut mining and power station fallout were studied using the Environmental Distress Scale (EDS) and interviews. It was found that the transformation of the landscape directly triggered feelings of solastalgia. Study participants experienced negative physical repercussions from the mining, such as respiratory issues, cancer, and weight loss whilst also experiencing negative emotional effects. Albrecht et al. [22] assert that for residents of the Upper Hunter, their sense of place, their identity, physical and mental health, and general wellbeing were all challenged by unwanted change. Consequently, they felt powerless to influence the outcome of the change process. This research is useful in forging a connection between physical devastation and identity.

The limitations of that particular study align with the majority of solastalgia-related research in that it did not describe the socio-political landscape. Members of the Upper Hunter community who were represented in the article were not Indigenous to the places they lived and as such, their connection came from working the land or from a generational connection to specific places. This is distinct from the understandings of place and land that Indigenous, and in light of the focus of this discussion, Pacific peoples may hold. Though this tension is acknowledged by Albrecht in his initial examination of the term solastalgia [7], this is unfortunately absent from research using this notion. The research is also focused on human-made changes to the environment. Though there is mention of climate change, as noted, this is not the main focus of the Environmental Distress Scale (EDS) questions or the conceptual framework. 

Building on the use of solastalgia as applied to human-made changes to the environment, Warsini, Mills, and Usher [23] theorised on the potential use of solastalgia for assessing the wellbeing of natural disaster survivors such as the effects of volcanic eruptions. In line with Albrecht, they emphasized that the detrimental alteration of ecosystems has the potential to undermine people’s sense of identity, belonging, and agency, leading to feelings of distress, sorrow, and despair when they come to realize that the cherished places they once inhabited have been permanently scarred and altered. They argue that solastalgia can be applied to natural disaster research as well as affirming the usefulness of such a term in future intervention development for health professionals. 

### 3.5. Solastalgia and Climate Change

There is growing scholarship on solastalgia and climate change. Cunsolo and Ellis [24] discussed the need for further research and theoretical development to capture the growing concern about ecological grief caused by climate change. They drew connections between the ecological grief experienced by Inuit communities in Canada and Ellis and Albrecht’s [25] research pertaining to Australian family farming communities. Cunsolo and Ellis emphasize that it is not just physical devastation, but rather the grief caused by the associated loss experienced by individuals extends beyond their personal knowledge and identity and is linked to their deep connection with the land. It also encompasses the erosion of a cultural framework of land-based knowledge that has been passed down through generations. This is an important expansion in solastalgia-related research as it introduces other terminology and affirms the differential experiences of place-based identity for Pacific peoples. Terms such as ecological grief may prove to hold more utility for climate change and mental health distress research; however, there remains a desire to understand the key drivers and risk factors associated with ecological grief. Caution should be applied when applying ecological grief, as the degree to which ecosystems and landscapes are considered distinct ‘places’ remains unclear, and the existing uncertainties in defining and conceptualizing crucial place-related notions, such as place attachment, place identity, and solastalgia, increase the likelihood of ecological grief being confused with related concepts [22]. 

In Ellis and Albrecht’s [25] case study on family farmers in the Australian Wheatbelt, they claim that a sense of place is not only a foundational concept for Indigenous peoples but also for those who hold close ties to the land through their work and livelihoods (e.g., farmers). Using a set of interviews with 22 farming families, they concluded, that participants felt a strong sense of place identity with their farmland; several participants drew parallels between their relationship to the land and likened this to Aboriginal Australians’ sense of place and belonging. This underscores a consistent limitation of solastalgia research and its inability to draw on driving factors of ecological distress/grief through the absence of a socio-political lens. This is useful in a Pacific context for examining subsistence lifestyles. Furthermore, it highlights the significance of *place* in solastalgia research, which differs from other terms for environmental distress that do not always relate to a sense of place so directly. However, this is also potentially restrictive for Pacific contexts. As Galway et al. [9] highlight, solastalgia studies do not endeavour to define what land means to particular people. Concepts of place and place identity are highly contextual and dependent upon who the ‘stakeholders’ are and the type of ‘stakes’ they hold to a particular place. In Ellis and Albrecht’s [25] case study, farmers hold different stakes in land than Indigenous peoples. Land is closely tied to their lifestyle and livelihoods as well as their deep personal and familial narratives intertwined with the land over extended periods of time. However, the parallels between farmers and Indigenous peoples in Ellis and Albrecht’s [25] paper are tenuous, given Indigenous peoples already experience displacement. This draws upon the settler politics of the land in Australia as well as the commodification of land. Therefore, the relationship to land and the manifestation of place identity is foundationally different for Indigenous and Pacific populations, despite the presence of expressions of grief from both Pacific and Western worldviews.

### 3.6. Place Identity

In relation to the previous discussion, it is further supported by Fresque-Baxter and Armitage [26], who argue for the importance of understanding place identity in climate change adaptation measures. They call for the linking together of an examination of place identity and climate change adaptation using an integrated approach that comprises behavioural intentions, health/wellbeing, and collective action as a critical pathway for comprehensive research and practical endeavours. This entails investigating the role of place identity theory as a framework to systematically explore how individuals’ connections with their identified places can contribute to understanding adaptation and adaptive capacity, both in theoretical understanding and real-world application.

It will be important to ask how place identity, and the formation of a specific place identity in the Pacific region, can better determine appropriate adaptation measures. Moreover, incorporating place identity theory may also lend itself to better socio-political analysis within climate change and mental health research. 

### 3.7. Practical Applications of Solastalgia

There appear to be limited ‘practical’ applications of the term solastalgia in the literature; however, Kennedy [27] notes that solastalgia may have some judicial weighting. In her paper outlining the case of the Bulga community, who successfully overturned an application to expand open-cut mining in their community, Kennedy details the use of solastalgia evidence submitted before the court by Albrecht. Solastalgia and Albrecht’s research was utilised to support the Progress Association’s assertion that changes to the physical environment, due to an expansion in open-cut mining, would significantly modify the community’s structure and intensify the existing sense of place loss experienced. Whilst there may be some hesitation around the nature of the research and how it was framed and conducted, it is still useful in accentuating some of the social cost that comes with environmental destruction. As Kennedy maintains, solastalgia highlighted the substantial evidence indicating that the community of Bulga would experience sizeable changes, leading to significant psychological distress among its members. Evidently, solastalgia, in this example, as a concept and as an applied tool of research, was useful in drawing attention to the experiences of stress and anxiety caused by environmental transformation. This may be particularly beneficial when we consider the potential for social policy change in light of climate change. 

## 4. Conclusions

This viewpoint has explored the conceptual term solastalgia and assessed its meaningfulness in Pacific-related climate change research. It is clear that solastalgia research has proven useful for particular contexts: mostly for settler communities that rely on their environment and land for economic and social livelihoods. Whilst there has been some engagement with solastalgia and Indigenous communities, this remains a minor area of solastalgia-focused work and further, the Pacific is invisible in this discourse. There is no denying that solastalgia has made an impact, but what is clear is that more broad and holistic perspectives are needed. Evidently, there is a need for future research to delve into the psychological and mental wellbeing aspects of climate change for underserved or marginalized communities as a focus within environmental and climate-related research. In particular, to contend the boundaries of what place identity and mental health distress may constitute. The lack of research conducted by, with, and for Pacific peoples underscores that this Western-framed concept may not entirely align with their worldviews as it is limited in its acknowledgement of their deep and ancestral connections to the land. It is imperative, that further research in the Pacific will need to consider both local and global contexts, in addition to placing current climate-related distress within longer historical narratives of mobility, displacement, adaptation and migration. Our *Climate Change and Mental Wellbeing: The Impact on Pacific Peoples* study investigating the links between climate change and wellbeing in Niue, the Cook Islands, and Aotearoa New Zealand is examining this further. All three sites have long histories of environmental degradation not only at the hands of climate change but equally, as a result of development, colonisation, settler colonisation, and globalisation, and as such, the study provides a unique opportunity to bridge the gap between current concepts of solastalgia and actual realities of climate distress for Pacific communities impacted by climate change.

## Figures and Tables

**Table 1 ijerph-20-07041-t001:** Tenets of Solastalgia.

Tenets of Solastalgia
- Can be caused by natural and artificial environmental change.
- Does not necessarily relate to the past—may seek “alleviation in a future that has to be designed and created” [7] (p. 45).
- Relates to any context where place identity is challenged [7].
- “Universal relevance in any context where there is the direct experience of transformation or destruction of the physical environment (home) by forces that undermine a personal and community sense of identity and control” [7] (p. 46).
- Can be viewed as a philosophical or psychosomatic illness (or both).

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
