# Peer review of "Is the Concept of Solastalgia Meaningful to Pacific Communities Experiencing Mental Health Distress Due to Climate Change? An Initial Exploration"

_ijerph, 2023, doi:10.3390/ijerph20227041_

Round 1
Reviewer 1 Report
Comments and Suggestions for Authors
This is a very important piece that helps in the decolonising literature on climate change. It is a foundational piece. I would love to see more research built from this paper on a term that resonates more with us as Indigenous and Pacific people and further move forward the decolonising of Climate Change Research.
In my view it only needs one more proof read for better flow (minor), then its good to go.
Fa'afetai tele lava ma ia manuia.
Fa'afetai tele lava.
Author Response
We would like to thank the reviewer for their invaluable time and their extremely encouraging comments. We agree, that this work is foundational and can confirm that an impending work based on this current piece, is in progress. We can also confirm, that we have proof read the manuscript and addressed minor edits, although not stipulated by the reviewer where those minor edits could be, we have addressed the obvious and consider the paper flows much better.
Our many thanks again to the reviewer for their strong and positive comments.
Reviewer 2 Report
Comments and Suggestions for Authors
A review of “Is the concept of solastalgia meaningful to indigenous and Pacific communities experiencing the impacts of climate change?”
The title of this paper says that the focus will be about climate change and the Pacific people in connection to a shared distressful or alienating experience. The authors conclude that the concept of solastalgia is a mixed bag to understand Indigenous experience. This may be so. However, reading this paper did not help me better understand the meaningfulness of examining the question in the title. As an academic paper, there is a number of shortcomings, too. First, this paper needs to explain its methodology. In order to accomplish the four objectives the authors laid out in the introduction, what methods and analyses did they use? Second, the title includes “Indigenous” but the four objectives mention only “Pacific populations.” Third, much of the discussion in the main body sounds like a literature review, which does not focus on the Pacific. Detailed discussions are about Canada and Australia. Fourth, I have some issues about the tendency of generalizing Indigenous peoples’ climate change experience. Overall, as far as I am concerned, this paper is not readable and I ask for a major revisions at least.
Title
Keywords include “mental health” but the title does not have it. Keywords do not include solastalgia. Let’s clarify the focus in this paper.
Abstract
It needs to explain about the methodology.
Introduction
Pages 2-3: Discussion about solastalgia sounds redundant.
-: “Of particular importance was losing connection to ancestors and history was, as well as nurturing of resilience factors unique to Pacific peoples such as connection to culture and family. [NZ Health Research Council]
Aim: “to introduce the discussion on solastalgia in the Pacific by; 1. Defining solastalgia; 2. Investigating the use of solastalgia in Pacific research; 3. Outlining the critiques and limitations of solastalgia; and 4. Exploring the appropriateness of this term for Indigenous and Pacific populations.
Methodology is missing in this paper.
What sources did the authors use? Do they have any specific Pacific Islander society or societies in mind? A large portion of this paper is dedicated to discuss past studies in regions other than the Pacific, including Australia, Canada, Ghana. Also, I wonder if it is possible to generalize the experience among the Pacific peoples? If so, explain how. Also, as the authors often refer to their project, “climate change and mental wellbeing,” explain what this is.
Solastalgia in Research
-: “Hunter has highlighted the compounding effect that climate change will pose for the mental wellbeing of Indigenous peoples in Australia, including Torres Strait islanders, who already experience adverse mental health challenges and outcomes. (4)
-: “Pacific peoples in Aotearoa New Zealand already experience higher rates of mental health challenges compared to the compared to the general population. (4)
Limitations of solastalgia
Page 4, lines 161-166: Here the discussion is vague and difficult to understand. Is this discussion about literature review? Did some past study mention about “potential solastalgia research on climate change and mental health”? If so, cite the source.
Page 4, lines 167-168: Explain how the concept of solastalgia help climate change policy development. Local adaptation policy development or national policy?
Page 4, 169-170: Explain why solastalgia does not speak to displacement. Displacement is one of major climate change consequences.
Page 4, 172-175: Is solastalgia about individual experience or collective experience? A western understanding of place is a collective notion. Here both individual and collective notions are intermixed. This needs further explanation.
Page 5, lines 220-231: I found the discussion in this paragraph dubious. In northern Ghana, there are multiple ethnic communities along with new migrant communities. Migration in this discussion sounds to be connected to urban migration; if so, those who moved to cities like Accura or Kumasi did not always move there due to community experience like weather changes, disasters, and food insecurity. The point about “multi-layered” is correct but not so about distress. Many who migrated from the north tend to work for stressful street-side vending with meager income. Some studies found that smallholders in northern Ghana moved to cities largely due to their high expectations for education, welfare, and other urban conveniences along with remittances. Another point here is that the discussion here overlaps with the next subsection, “Climate Induced Migration and Mental Health.” This means a need to more carefully organize the discussion in this paper.
Comments on the Quality of English Language
Poor essay organization in this paper. Some subsections are redundant.
Author Response
We would like to thank for the reviewer for their invaluable insights and time given to review our manuscript. Please refer to file outlining point-by-point responses.

Reviewer 3 Report
Comments and Suggestions for Authors
I recommend publication of the manuscript with minor revisions. The paper is well written, arguments are clear and have an orderly flow. I have two main comments. I would suggest the authors expand or broaden the scope of their argument and make more emphasis on the importance of understanding or exploring the mental/phycological consequences of climate change -beyond just the use of the term solastalgia. Secondly,I suggest the authors engage further with the wealth of literature on conflict, environment and community healing- a few suggestions are included at the end of my comments.
The authors present convincing arguments to substantiate the claim that the concept of solastalgia falls short in explaining the possible mental health issues faced by indigenous people and people of the pacific. It is not surprising since one cannot expect to explain complex interactions and outcomes within complex social systems and between social-ecological systems using a single concept or approach- any one term or concept would imply an obvious oversimplification. In my view the authors make this point strong and clear in the first few pages. They also present arguments for a broader more universal case; mental and psychological implications of rapid and drastic environmental change are often neglected or addressed with oversimplified, often short term solutions. I would suggest the authors highlight their arguments on the unique pressures posed on indigenous people and peoples of the pacific by climate change. I would recommend not focusing on expanding the term solastalgia (perhaps just discard it all together since it seems from the authors perspective to be very limited to begin with) but rather encourage a more holistic approach to understanding human-nature relationships and conflicts. One that includes a psychic/ psychological/ mental dimension. The point of the manuscript should then be slightly broadened as to avoid presenting a direct attack on one term and rather emphasize an invitation to acknowledge the mental and psychological dimensions of climate change in environmental or climate related research in particular within marginalized or more vulnerable communities. This would perhaps require the authors to rewrite some of the last sections mainly : Practical Applications of solastalgia and Expansions of solastalgia as well as some paragraphs in the conclusion section.
In addition I would suggest the authors expand their references and engage with the wealth of literature that already exists on conflict and environment and also on community woundedness and healing (see a few examples bellow)
On conflict and environment
Castro, A. P., & Nielsen, E. (2001). Indigenous people and co-management: Implications for conflict management. Environmental Science & Policy, 4(4), 229–239. https://doi.org/10.1016/S1462-9011(01)00022-3
Galaz, V. (2005). Social-ecological Resilience and Social Conflict: Institutions and Strategic Adaptation in Swedish Water Management. AMBIO: A Journal of the Human Environment, 34(7), 567–572. https://doi.org/10.1579/0044-7447-34.7.567
Goldstone, J. A. (2018). Demography, environment, and security. In P. F. Diehl & N. P. Gleditsch (Eds.), Environmental Conflict (pp. 84–108). Routledge.
O’Riordan, M., Mahon, M., & McDonagh, J. (2015). Power, discourse and participation in nature conflicts: The case of turf cutters in the governance of Ireland’s raised bog designations. Journal of Environmental Policy & Planning, 17(1), 127–145.
Pellizzoni, L. (2011). The politics of facts: Local environmental conflicts and expertise. Environmental Politics, 20(6), 765–785. https://doi.org/10.1080/09644016.2011.617164
Str√¶de, S., & Helles, F. (2000). Park-people conflict resolution in Royal Chitwan National Park, Nepal: Buying time at high cost? Environmental Conservation, 27(4), 368–381.
Van Assche, K., Gruezmacher, M., & Beunen, R. (2022). Shock and Conflict in Social-Ecological Systems: Implications for Environmental Governance. Sustainability, 14(2), 610. https://doi.org/10.3390/su14020610
On community woundedness and healing
Chioneso, N. A., Hunter, C. D., Gobin, R. L., McNeil Smith, S., Mendenhall, R., & Neville, H. A. (2020). Community Healing and Resistance Through Storytelling: A Framework to Address Racial Trauma in Africana Communities. Journal of Black Psychology, 46(2–3), 95–121. https://doi.org/10.1177/0095798420929468
Erikson, K. (1994). A New Species of Trouble: Explorations In Disaster, Trauma, and Community. Norton.
Freire, P. (2018). Pedagogy of the Oppressed: 50th Anniversary Edition. Bloomsbury Academic.
Renes, H. (2022). Landscapes of Conflict and Trauma. In H. Renes, Landscape, Heritage and National Identity in Modern Europe (pp. 63–69). Springer International Publishing. https://doi.org/10.1007/978-3-031-09536-8_7
Van Assche, K., Gruezmacher, M., & Granzow, M. (2021). From trauma to fantasy and policy. The past in the futures of mining communities; the case of Crowsnest Pass, Alberta. Resources Policy, 72, 102050–102050. https://doi.org/10.1016/j.resourpol.2021.102050
van der Watt, P. (2018). Community development in wounded communities: Seductive schemes or un-veiling and healing? Community Development Journal, 53(4), 714–731. https://doi.org/10.1093/cdj/bsx017
Author Response
|
We would like to convey our appreciation to the reviewer for their time and invaluable expertise. We thank the reviewer for their opening statement: “I recommend publication of the manuscript with minor revisions.” We would also like to thank the reviewer for their positive comment - “The paper is well written, arguments are clear and have an orderly flow.” The reviewer proposes two suggestions:
|
|
We thank the reviewer for affirming our position whereby - “The authors present convincing arguments to substantiate the claim that the concept of solastalgia falls short in explaining the possible mental health issues faced by indigenous people and people of the pacific [sic].” In addition, we are pleased the reviewer recognizes our broader arguments, by which, mental health implications are often neglected or oversimplified. Again, we appreciate the suggestions made by the reviewer. As mentioned, regarding an impending follow up piece, we will be focusing more specifically on the examination of Pacific peoples’ perspectives of the mental/psychological impacts of climate change which will expand upon the term solastalgia and its appropriateness (or not) in Pacific context sand further highlight the need for alternative articulations from Pacific peoples themselves. We have, in the interim, added a statement being more explicit about this upcoming work (see lines 100-103). We have reviewed the last sections of our paper, and amended mainly, ‘Practical Applications of Solastagia’ and ‘Expansions of Solastagia’ as suggested by the reviewer. We thank the reviewer for their suggestion to discard the ‘expansion on solastalgia’ section. We agree and have discarthis section accordingly.We have includied a statement emphasizing that more holistc approaches is needed (see lines 438-439). We have tweaked slightly, statements made under the ‘Practical Applications of Solastalgia’ in response to the reviewer’s comment and have softened the tone (amendments highlighted). We also note the reviewer’s comment in emphasizing the “mental and psychological dimensions of climate change in environmental or climate related research in particular within marginalized or more vulnerable communities.” We have included this statement (see lines 439-446). |